# Alcohol-Related Liver Disease: An Overview on Pathophysiology, Diagnosis and Therapeutic Perspectives

**DOI:** 10.3390/biomedicines10102530

**Published:** 2022-10-10

**Authors:** Yoonji Ha, Inju Jeong, Tae Hyun Kim

**Affiliations:** 1College of Pharmacy, Sookmyung Women’s University, Seoul 04310, Korea; 2Drug Information Research Institute, Sookmyung Women’s University, Seoul 04310, Korea; 3Muscle Physiome Research Center, Sookmyung Women’s University, Seoul 04310, Korea

**Keywords:** alcohol-related liver disease, pathophysiology, diagnosis, non-alcoholic fatty liver disease, therapeutic targets, G protein-coupled receptor

## Abstract

Alcohol-related liver disease (ALD) refers to a spectrum of liver manifestations ranging from fatty liver diseases, steatohepatitis, and fibrosis/cirrhosis with chronic inflammation primarily due to excessive alcohol use. Currently, ALD is considered as one of the most prevalent causes of liver disease-associated mortality worldwide. Although the pathogenesis of ALD has been intensively investigated, the present understanding of its biomarkers in the context of early clinical diagnosis is not complete, and novel therapeutic targets that can significantly alleviate advanced forms of ALD are limited. While alcohol abstinence remains the primary therapeutic intervention for managing ALD, there are currently no approved medications for treating ALD. Furthermore, given the similarities and the differences between ALD and non-alcoholic fatty liver disease in terms of disease progression and underlying molecular mechanisms, numerous studies have demonstrated that many therapeutic interventions targeting several signaling pathways, including oxidative stress, inflammatory response, hormonal regulation, and hepatocyte death play a significant role in ALD treatment. Therefore, in this review, we summarized several key molecular targets and their modes of action in ALD progression. We also described the updated therapeutic options for ALD management with a particular emphasis on potentially novel signaling pathways.

## 1. Introduction

Alcohol-related liver disease (ALD) is one of the most common hepatic manifestations primarily caused by excess alcohol consumption. ALD is one of the most predominant causes of liver disease-related morbidity and mortality worldwide, especially in Europe and the USA [1,2,3]. According to a recent report from the World Health Organization (WHO, Geneva, Switzerland), the adult per capita alcohol intake has globally increased by over 10% in the past 25 years, contributing to an increase in the burden of alcohol-related disease with approximately 3 million deaths and 132.6 million disability-adjusted life years (DALYs) worldwide in 2016 [4,5]. Similar to other hepatotoxicants, the severity of ALD is highly dependent on the amount and duration of alcohol intake and the drinking pattern (chronic or binge), which varies considerably in each individual. The US Department of Agriculture Dietary Guidelines proposed a definition of moderate alcohol intake as up to 2 drinks per day for men and 1 drink for women (a standard drink contains approximately 12.5 g of ethanol). Excess consumption of >40 g of pure alcohol per day over a prolonged period (years) usually serves as a primary risk factor for developing ALD [5,6]. Moreover, another recent study has revealed that the chronic intake of alcohol at approximately 12–24 g per day can also lead to alcoholic cirrhosis [5,7], suggesting a high cumulative risk of alcohol consumption even at a comparably low threshold level in ALD. Meanwhile, binge drinking, which shows an entirely different pattern of alcohol consumption, also posits another common and deadly form of excessive alcohol intake in Western countries. According to the National Institute on Alcohol Abuse and Alcoholism (NIAAA), binge drinking is defined as the consumption of five or more drinks for men or four or more drinks for women in about 2 h; however, it is implausible to estimate the exact amount of alcohol intake per binge drinking and also it may not take into account the possibility of having multiple binges during a short period of time (e.g., within several hours or a day) [8,9]. Individuals with high or sustained alcohol intake are at increased risk of ALD, and approximately 10–20% of these patients further develop alcoholic cirrhosis [10]. Notably, recovery from alcohol-induced liver damage highly depends on abstinence from alcohol [5,9,11].

## 2. Pathophysiology of ALD

### 2.1. Alcohol Metabolism

Alcohol, being soluble in both water and lipid, can easily diffuse across cell membranes. After being absorbed into the circulation from the gastrointestinal tract, alcohol is primarily metabolized by the liver; however, only a small amount ≤ 10% is directly eliminated via lungs, kidneys, and sweat in its intact form [11,12]. The enzymatic metabolism of alcohol which involves oxidative biotransformation into acetaldehyde is primarily mediated by alcohol dehydrogenase (ADH) [11,13]. ADH is expressed predominantly in the liver and to a much lesser extent in the gastrointestinal tract, where it catalyzes the oxidation of alcohol into acetaldehyde using cellular nicotinamide adenine dinucleotide (NAD^+^) as a co-factor [11,13]. As a highly reactive, toxic metabolite of alcohol, acetaldehyde is metabolized by acetaldehyde dehydrogenase (ALDH) into acetate, and afterward excreted from hepatocytes into the bloodstream. Amongst the twelve ALDH genes identified in humans, *ALDH2* is a conserved mitochondrial enzyme mainly expressed in hepatocytes and notably implicated in the detoxification of acetaldehyde [11,13,14]. ALDH2 is of interest in cellular metabolism due to its critical role in oxidizing lipid peroxidation products. In addition, a substantial population of East Asians is at increased risk of acetaldehyde accumulation due to the inherited *ALDH2*2* allele (non-functional ALDH2 gene), causing alcohol flushing syndrome, including facial flushing, tachycardia, nausea, and headache [14]. Although the ADH pathway primarily takes charge of alcohol metabolism when the level of alcohol concentration in the body is low, supporting systems also exist when the concentration of alcohol is higher beyond the ADH metabolic capacity. These include: (a) Microsomal ethanol oxidation system (MEOS), in which cytochrome P450 2E1 (CYP2E1) is considered the most critical component and also posits an important role in alcohol oxidation [11,13]. Under normal physiological situations, CYP2E1 accounts for 10% of alcohol oxidation, as the ADH-ALDH system is generally responsible for metabolizing alcohol in hepatocytes [11,15]. However, CYP2E1 expression is strongly induced, particularly near perivenous regions (zone 3), in response to excess alcohol exposure, which partially explains the differential patterns of injury across hepatic lobules [11,13,15,16]. Moreover, overexpressed CYP2E1 is closely associated with alcohol-induced liver injury due to enhanced reactive oxygen species (ROS) generation during the enzymatic process, resulting in increased oxidative stress and inflammatory response [5,11,15]. (b) Catalase, a hydrogen-scavenging enzyme mainly localized in the peroxisome, also participates in the oxidation of alcohol, even though its contribution to alcohol metabolism is generally insignificant in the liver under normal conditions [13]. However, it has been reported that catalase-mediated alcohol oxidation becomes the major pathway in the fasting state [17], the importance of which is further pronounced in the brain where ADH is not expressed [18,19]. In addition, a recent study also showed that the activation of the peroxisome proliferator-activated receptor α (PPARα)-catalase pathway ameliorates alcoholic liver injury via increasing NAD^+^ synthesis and accelerating alcohol clearance [19], thus supporting the key role of catalase in alcohol metabolism.

### 2.2. ALD Progression

#### 2.2.1. Alcoholic Fatty Liver (Steatosis)

Hepatic fat content can be determined as a net consequence of the interplay between underlying lipid metabolism-associated pathways (e.g., fatty acid synthesis (de novo lipogenesis) and triglyceride accumulation, the influx of free fatty acids/chylomicrons from extrahepatic tissues to the liver, the efflux of triglyceride/cholesterol from the liver to the circulation, and mitochondrial fatty acid oxidation) [5,20]. Although alcohol and acetaldehyde do not directly serve as building blocks for fatty acid synthesis, sustained alcohol consumption results in fat accumulation in hepatocytes through several mechanisms resulting from deregulated lipid metabolism, leading to alcoholic fatty liver (AFL) [11,20,21]. The alcohol oxidation process consumes and thus disrupts the NAD^+^-NADH redox potential, causing suppression of mitochondrial fatty acid oxidation and promoting lipogenesis [5,11,20]. In line with this, it has been reported that alcohol increases the expression of sterol-regulatory element binding protein 1c (SREBP-1c), a master regulator of lipogenic genes, whereas it downregulates *PPARα* gene expression, a key transcription factor that enhances fatty acid oxidation [5,11,22,23]. Moreover, the opposing regulations of SREBP-1c and *PPARα* expression by alcohol can be readily explained by the downregulation of AMPK (AMP-activated protein kinase; a master regulator of energy metabolism), which is largely accompanied by the induction of endoplasmic reticulum stress (ER stress) and ROS accumulation [11,24,25]. Collectively, all of these mechanisms promote disease progression toward AFL.

#### 2.2.2. Alcoholic Steatohepatitis (ASH)/Alcoholic Hepatitis (AH)

Sustained exposure to alcohol in hepatic steatosis may trigger an inflammatory response, ROS accumulation, and hepatocyte damage; thus, leading to alcoholic steatohepatitis (ASH), a prerequisite for progression to fibrosis, cirrhosis, and hepatocellular carcinoma (HCC). A relatively small proportion of heavy drinkers can develop alcoholic hepatitis (AH), which has a poor clinical outcome (i.e., 20–50% mortality within 3 months) and is associated with decompensated hepatic functions, abrupt jaundice, and ductular formation due to severe inflammatory response and hepatocellular damages [5,26,27,28]. AH is primarily characterized by its fast progression toward fibrosis, which may be responsible for its role as a major contributor to mortality and morbidity amongst patients with ALD in Europe and the USA [5,29,30]. Hepatic inflammation resulting from excessive alcohol consumption is characterized by hepatocellular injuries accompanied by inflammatory infiltrates mainly composed of polymorphonuclear leukocytes. The histological features of ASH include hepatocyte ballooning and Mallory–Denk body (i.e., a hepatic inclusion of aggregated cytokeratin) [20,26,31], which serve as characteristic markers when determining the grade and severity of ALD progression. In general, the severity of alcohol-driven hepatic inflammation is closely correlated with the circulating level of lipopolysaccharides (LPS), as it is one of the representative pathogen-associated molecular patterns (PAMPs) being recognized by pathogen-recognition receptors (e.g., Toll-like receptors), that induces inflammatory lesions in the liver [11,32,33]. Furthermore, an increase in proinflammatory cascade can provoke ROS generation via CYP2E1 and NADH-dependent cytochrome reductase, NADPH oxidases, endoplasmic reticulum (ER) stress response, and mitochondrial dysfunction. These cellular responses may exacerbate hepatocellular damage and subsequent release of damage-associated molecular patterns (DAMPs), further exacerbating immune cell stimulation and thus developing fibrosis/cirrhosis and cancer [26,34,35,36,37,38].

#### 2.2.3. Fibrosis/Cirrhosis

Patients with severe ASH who persistently consume alcohol can further progress to fibrosis due to repetitive wound-healing processes in response to chronic inflammation and hepatocellular damage [5,20]. Liver fibrosis, which serves as a prerequisite for the development of cirrhosis, is characterized by the massive accumulation of extracellular matrix (mainly composed of collagen) around pericentral and perisinusoidal regions (chicken-wire appearance) [5]. Although the molecular mechanisms of fibrosis in ALD are not completely defined, extracellular matrix production by activated hepatic stellate cells (HSCs), and to a lesser extent, by portal fibroblasts and myofibroblasts play key roles in fibrogenesis. HSCs can be directly activated by alcohol, acetaldehyde, and ROS themselves or in a paracrine manner by various fibrogenic factors (e.g., transforming growth factor-β, platelet-derived growth factor, and inflammatory cytokines) released from damaged hepatocytes, activated Kupffer cells, and infiltrated immune cells [5,20,39]. In line with this, it has been shown that alcohol suppresses the clearance of activated HSCs by natural killer (NK) cells, thereby aggravating fibrosis. Once this fibrogenic process becomes advanced and persistent, hepatic architecture is severely modified, and the lobular vasculature becomes narrowed, affecting the distribution of hepatic blood flow and may cause portal hypertension, ascites, and other complications [5,40,41].

### 2.3. ALD and Non-Alcoholic Fatty Liver Disease (NAFLD): Similarities and Differences

While ALD has been regarded as one of the most predominant liver diseases affecting approximately 2.5% of the general population over several decades, non-alcoholic fatty liver disease (NAFLD) has recently emerged as another major hepatic manifestation that can develop into severe forms of liver pathogenesis, which constitutes nearly 25% of liver-related morbidity worldwide [42,43,44,45,46,47]. In particular, non-alcoholic steatohepatitis (NASH) was firstly defined and described by Ludwig et al. in 1980 from their clinical observations, which showed elevated liver enzymes and comparable histopathological features with that of AH but without an apparent contribution of alcohol consumption [42,48]. Although the major metabolic determinant for each disease comes from totally different sources (ethanol for ALD and excess nutrient—mostly free fatty acids for NAFLD), NAFLD shares similar pathophysiology in many aspects with ALD throughout the typical spectrum of the disease. At the same time, distinguishable clinical and histological characteristics exist between the two diseases, given their markedly different etiologies. ALD and NAFLD have analogous histological outcomes such as mixed macro- and micro-vesicular steatosis, hepatocyte ballooning in conjunction with infiltration of inflammatory cells, and Mallory–Denk bodies [42,49,50,51]. These parallels largely result from similar pathogenesis, including deregulated lipid metabolism in hepatocytes (i.e., aggravated lipid accumulation in conjunction with suppressed fatty acid oxidation), organelle dysfunction (e.g., ER and mitochondrial stress), oxidative stress-inflicted cell death (e.g., extrinsic and intrinsic apoptosis), elevated inflammatory response (e.g., Kupffer cell polarization toward high M1/M2 ratio) and hepatic stellate cell-mediated fibrogenesis [42,51]. Further, both ALD and NAFLD share a genetic predisposition that contributes to their disease progression through hepatic fat accumulation (*PNPLA3, TM6SF2, GCKR, GPAM, APOB, PYGO1*, etc.), oxidative and ER stress (*HFE, MARC1, SOD2, UCP2, SERPINA1*), inflammation and fibrogenesis (*PNPLA3, HDS17B13, MERTK, LEPR*) [52]. While ALD and NAFLD exhibit largely conserved pathophysiological implications, several divergent molecular pathways have been identified: In macrophage, MyD88 is recruited to Toll-like receptor 4 (TLR4) upon ligand binding and then activates downstream signaling in NASH, while MyD88 seems not to be involved in TLR4 signaling in ASH [42,53,54]. Likewise, a recent finding suggests that ASH induces more severe ER stress than NASH following the enhanced expression of genes triggering apoptosis [55]. Recent studies have also demonstrated that in both ASH and NASH, several other forms of hepatocyte death are observed, including necroptosis, pyroptosis, and ferroptosis, which may partially explain why inhibition of apoptosis alone cannot completely alleviate hepatocyte damage [42,46]. In particular, necroptosis is found to play a significant role in facilitating the disease progression through a mixed-lineage kinase domain-like (MLKL)-receptor-interacting protein kinase (RIPK)-1/3 signaling [42,56]. ASH exhibits necroptosis-mediated cell death in RIPK-3-dependent and RIPK-1-independent manner [57], whereas RIPK-3 seems not to participate in NASH development [58], even though the role of RIPKs is still controversial [42,46,59]. Pyroptosis has recently received great attention as a newly identified form of programmed cell death in the context of innate immunity [60]. In both ASH and NASH, pyroptosis activates the inflammasome complex in response to proinflammatory signals, exhibiting the secretion of the interleukin (IL)-1 family as well as the continuous release of DAMPs which further exacerbates inflammatory response [42,46,60]. Meanwhile, there are some distinctions in inflammasome signaling between the two entities; in ASH, inflammasome activation typically occurs in macrophages and to a much lesser extent in hepatocytes [61,62], whereas it is usually observed in hepatocytes in NASH [63,64]. Furthermore, inflammasome activation and IL-1 secretion are observed in the early phases of ASH progression, which is mostly seen in the late stages of NASH [42,61,63]. There are also other several pathogenic mechanisms that can affect the disease characterization upon different dietary challenges between ASH and NASH, including changes in microbiome composition and circulating LPS level, diet-induced alterations in lipotoxicity, insulin resistance, and damaged hepatocytes-derived secretory factors (e.g., extracellular vesicle-contained chemokines and microRNAs, hepatokines, etc.) [42,65,66,67,68,69]. Since many patients with chronic liver disease frequently have a history of both heavy alcohol consumption and excess dietary caloric intake, understanding the similarities and the differences of these pathogenic mechanisms mentioned above would be essential and thus help in developing the therapeutic strategies against ASH and NASH (Table 1) [5,42,51].

## 3. Clinical Diagnosis of ALD

In general, the clinical symptoms of ALD do not appear until it develops toward moderate to advanced stages, making it difficult to diagnose ALD during the early phase. However, one of the most simple and convenient methods to detect the early stage of ALD is to perform a liver function test using blood biochemical parameters. The level of aminotransferases in blood samples usually reaches up to 5–8 times higher in ALD patients than that of normal individuals, and the ratio of aspartate transferase (AST) to alanine transferase (ALT) over 2 (i.e., AST/ALT > 2) is generally and distinctively observed in patients with ALD [91]. However, when it comes to verifying alcoholic origin in patients with cirrhosis, the sensitivity and feasibility of the AST/ALT ratio may be compromised [51,92]. Similarly, gamma-glutamyl transpeptidase (GGT) is also significantly elevated in the plasma of ALD patients as it serves as another biomarker for inflamed fatty liver after heavy alcohol consumption. However, it suffers from a lack of specificity since the elevated serum level of GGT is also found in patients with increased body mass index (BMI), cholestatic liver disease, drug-induced liver injury (DILI), cardiac insufficiency, and others [5,51,93,94]. Rather, the serum level of caspase-cleaved cytokeratin 18 (CK18-Asp396) can serve as a potentially useful biomarker compared to transaminases considering their higher sensitivity for detecting hepatocytes undergoing apoptosis in ALD [5,95]. The measurement of fibrosis is critical in assessing the degree of disease progression in advanced ALD. The extent of hepatic fibrosis largely correlates with the level of liver stiffness, which is commonly assessed by non-invasive elastography techniques (e.g., fibroscan, shear wave elastography, magnetic resonance elastography, etc.) [5]. Other fibrosis measurement tools for evaluating liver stiffness include the Fibrotest and Enhanced Liver Fibrosis (ELF) test, and both methods can be utilized together with the above-mentioned serum biomarkers for better interpretation and more accurate assessment [5,26,51].

## 4. Therapeutic Options for Various Stages of ALD

### 4.1. Lifestyle Modification and Nutritional Intervention

Although many researchers and clinicians have been making an enormous effort to identify novel therapeutic targets against ALD/AH, the cessation of alcohol would still be the most effective and safe intervention for the management of ALD/AH regardless of the extent of disease progression [5,9,11]. Notably, hepatic steatosis may be effectively reversed after sustained abstinence from alcohol, thus showing the critical impact of lifestyle modification and the importance of early detection. Nutritional intervention is widely recommended for patients with mild AH, such as high protein diets, vitamin B, C, K, and folic acid [26,96]. In particular, since many patients with severe AH are at high risk of malnutrition, the practice guidelines of the American Association for the Study of Liver Disease (AASLD) and the European Association for the Study of the Liver (EASL) both recommend a daily protein intake of ~1.5 g/kg body weight [5,20,91]. It was recently reported that everyday drinking up to two cups of coffee might have a potential benefit in lowering the risk of alcoholic cirrhosis [51,97]. In addition, the administration of B-complex vitamins is often advised for the prevention of Wernicke encephalopathy [20]. Since AH patients with poor prognoses are especially prone to severe infections, early and precise diagnosis with adequate medications (e.g., antibiotics) is required (Table 2).

### 4.2. Targeted Treatments

#### 4.2.1. Inflammation

Given the chronic pro-inflammatory state in patients with advanced forms of ALD [98], AASLD and EASL both proposed using corticosteroids for patients with severe AH based on their anti-inflammatory properties, which have shown a significant improvement in shortening 28-day mortality [5,99,100]. Recent studies have also revealed a reduction in short-term mortality of patients with severe AH, supporting the promising therapeutic efficacy of corticosteroids against alcohol-associated liver disease [101,102,103,104]. However, the limitation still exists as a substantial proportion of AH patients do not properly respond to conventional corticosteroid therapy, which may partially account for its poor performance in reducing 6-month mortality [28,99,103,104]. Therefore, careful consideration should be taken when administering corticosteroids in patients with high risks of infection, as this can greatly contribute to increased mortality rates [105,106,107,108]. Besides, EASL guidelines recommend not using corticosteroids in non-responders [5,99]. However, some pharmacological interventions have shown the potential to increase the responsiveness to corticosteroids, raising the necessity of developing novel therapeutic strategies [5,11]. A line of evidence has revealed that administration of antibodies directed against tumor-necrosis factor (TNF) showed promising outcomes in both alcohol-induced liver injury animal models and several pilot trials of patients with AH [109,110,111,112,113,114,115]. However, some other large randomized, controlled clinical studies evaluating etanercept or infliximab failed to confirm the initial results, rather they showed an increase in mortality rates [116,117], which might be attributed to the higher risk of infection by the sequential infusion of TNF antagonists and/or lowered hepatic regeneration capacity due to the blockade of TNF receptor [39,118]. As a result, pentoxifylline, a non-selective phosphodiesterase inhibitor, has been studied as another therapeutic option for alcoholic hepatitis based on, in part, its inhibitory effect on the synthesis of TNFα along with pro-inflammatory cytokines [119,120,121] and hepatorenal syndrome [11,119]. Combined treatment of pentoxifylline and corticosteroid has been examined in patients with AH, based on their different modes of action; however, the combination turned out to be ineffective in improving the survival rate of patients in a large randomized controlled trial [122]. Likewise, the use of pentoxifylline in combination with Anakinra, an IL-1 receptor antagonist, plus zinc showed similar survival benefits compared to methylprednisolone treatment in patients with severe AH [123]. Further, another large randomized controlled study (STOPAH) has demonstrated that patients treated with pentoxifylline showed no effect compared to the placebo group in terms of short-term mortality [11]. Hence, the current guidelines do not generally recommend the use of pentoxifylline alone to treat AH [91,99,124].

#### 4.2.2. Oxidative Stress

Given the importance and significance of oxidative stress in ALD progression, some antioxidant molecules that may enhance glutathione level have been initially considered promising therapeutic targets for the early stages of ALD [28,125,126]. In ethanol-feeding rodent models, administration of either S-adenosylmethionine or betaine showed a marked reduction in steatosis and mitochondrial damage [127,128]; however, the evidence for their effectiveness in clinical studies is yet limited [11]. Growing evidence has indicated that using antioxidant molecules alone may not be effective in improving the advanced forms of ALD [11,129,130,131]. Rather, it might be considered that some antioxidant molecules, such as *N*-acetylcysteine, can be used as preventive or prospective agents along with conventional medications. However, more evidence in human studies and molecular insights into their mechanisms of action are required [127,128,132]. Metadoxine, a hepatoprotective medicine with pyridoxine-pyrrolidone carboxylate moiety, has also been used to treat acute and chronic alcohol intoxication and has shown an improvement in reducing both short- and long-term mortality in AH patients when used in combination with either steroids or pentoxifylline [133,134,135,136]. Furthermore, several experimental animal models have revealed that numerous genes associated with cellular antioxidative defense mechanisms have been proposed as potentially promising targets for the treatment of ALD [26,137,138,139,140]. Interestingly, recent studies have shown that inhibition of CYP2E1, a key enzyme responsible for ROS generation during alcohol intoxication, by either direct pharmacological intervention or inverse agonism of upstream regulatory molecules significantly abrogated alcohol-induced oxidative liver injury in mice [141,142,143,144], suggestive of the potential for novel drug development against various stages of ALD.

#### 4.2.3. Hepatocellular Death

Hepatocytes heavily exposed to alcohol are highly susceptible to cellular damage inflicted by exacerbated oxidative stress and subsequent inflammatory response associated with ER stress, mitochondrial dysfunction, and caspase-dependent signaling pathway. Apoptosis is the most predominant form of hepatocellular death in ALD, while other types of cell death have also been reported to be significantly involved [145]. Given the key role of caspase family proteins in apoptosis, the use of emricasan (IDN-6556), a pan-caspase inhibitor, has been proposed for treating ALD based on its efficacy in several studies [136,146,147]. However, a recent clinical trial evaluating the efficacy of emricasan in patients with severe AH has been terminated due to the serious concern of high systemic drug levels that might have resulted from deregulated hepatic metabolism in those patients [148]. Similarly, selonsertib (GS-4997), an inhibitor of apoptosis signal-regulating kinase-1 (ASK1), has also been examined as an adjuvant medication to prednisolone, but it turned out that selonsertib has no benefit in 28-day mortality of patients with severe AH [136,149]. As apoptotic cell death is largely regulated by the counterbalance between pro-apoptotic and anti-apoptotic signaling pathways, many proteins can be proposed as potential therapeutic targets against various stages of ALD [20,26].

#### 4.2.4. Hepatic Regeneration

Liver regeneration can counteract and thus compensate against hepatocellular death which may alleviate ALD progression. The regenerative capacity of hepatocytes can be enhanced by increasing the production and subsequent release of stem cells into the circulation from bone marrow as mediated by the granulocyte-colony stimulating factor (G- CSF) [150]. Several clinical trials have revealed that G-CSF shows promising efficacy in reducing mortality as well as the risk of infection in patients with severe AH; however, more investigations in human studies may be necessary [151,152,153,154,155,156]. Similarly, IL-22, which is primarily produced by T helper type 17 cells and NK cells, has been ascribed as a promising target for the treatment of ALD based on its stimulatory effect on hepatoprotection and tissue repair (i.e., proliferation). Moreover, IL-22 also plays a key role in inhibiting bacterial infection, thereby serving as a possible therapeutic option that may alleviate the risk of corticosteroid-mediated infection [26].

Following the observation of the beneficial effects of IL-22 on inflammation and impaired hepatic regeneration in the ALD rodent model [157,158], F-652, a recombinant fusion protein of human IL-22, has proven its safety as well as efficacy in reducing inflammation while inducing hepatic regeneration in patients with moderate and severe AH [159]. Furthermore, obeticholic acid, a semi-synthetic agonist of farnesoid X receptor (FXR), has been shown to affect bile acid abnormalities, improve cholestasis, and promote liver regeneration [150,160]. FXR has also been established to play a key role in ameliorating portal hypertension [161], primary biliary cholangitis [162], and NAFLD [163]. Moreover, FXR agonism also showed enhanced gut barrier function in the rodent model of ALD, suggesting its possible therapeutic efficacy in individuals with ALD or AH [150]. However, the clinical trial conducted using obeticholic acid in patients with moderately severe AH was terminated due to hepatotoxicity issues [136,160].

### 4.3. Liver Transplantation

For subpopulations of patients with ALD who do not properly respond to medical therapies, early liver transplantation (typically before 6 months of alcohol abstinence) can be another potential therapeutic option, as supported by results from some clinical studies demonstrating remarkable improvement in the survival of patients with severe AH or alcoholic cirrhosis [28,164,165,166]. Nevertheless, proper managements, such as alcohol cessation, are still essential after liver transplantation for a successful outcome as continued alcohol consumption after the surgery is strongly associated with increased mortality [164,165,166]. Overall, although the numerous pharmacological and/or surgical therapies for ALD have been continuously and intensively explored during the past decades, permanent abstinence from alcohol is still considered the most critical intervention as a prerequisite management, raising the necessity of developing a novel therapeutic strategy for ALD.

### 4.4. Potential Novel Therapeutic Targets

#### 4.4.1. MicroRNAs

MicroRNAs (miRNAs) are short non-coding RNAs with generally 20–22 nucleotides long that play a crucial role in several cellular processes, including development, differentiation, immune response, energy metabolism, tumorigenesis, and more [167,168,169]. miRNAs that have undergone maturation processes can modulate the expression as well as the function of their target genes through post-transcriptional regulations. Moreover, a single miRNA is generally capable of regulating hundreds of genes [167], thus implicating the significant role of miRNAs in various diseases. Notably, recent studies have revealed that alcohol can affect miRNAs which significantly alters cellular responses to oxidative stress, inflammation, and organelle dysfunction [169,170]. To date, several miRNAs, including miR-155, miR-182, miR-132, and miR-34a are increased, while another subset of miRNAs such as miR-122, miR-148a, and miR-203 are decreased across a spectrum of ALD [26,136,169,170]. Emerging evidence from numerous studies using both animal and human liver specimens indicates that miRNAs can serve as promising therapeutic targets for ALD. However, as no ongoing clinical trials evaluating the efficacy of miRNAs in ALD treatment, in-depth investigations on the in vivo validation for therapeutic efficacy of those miRNAs, as well as their molecular insights into the mechanism of action, are required.

#### 4.4.2. Gut-Liver Axis

Chronic alcohol consumption increases bacterial overgrowth and dysbiosis by increasing the total intestinal bacterial burden and changing the composition of existing microbiota, leading to an increase in gut permeability and translocation of intestinal microbiome-derived pathogen-associated molecular patterns (PAMPs), including lipopolysaccharide (LPS) and bacterial DNA into the portal circulation [5]. PAMPs, together with DAMPs derived from hepatocytes, undergo apoptosis upon heavy alcohol exposure, then they strongly activate both innate and adaptive immune responses which cause excessive production and secretion of pro-inflammatory cytokines and chemokines, thus exacerbating inflammation. Due to this higher risk of infection and hepatic inflammation, several antibiotics (e.g., amoxicillin clavulanate, rifaximin) and probiotics (e.g., *Bifidobacterium bifidum*, *Lactobacillus plantarum 8PA3*) are under investigation in clinical trials for their therapeutic potential against AH [11,136,146,171]. Additionally, fecal microbiota transplantation (FMT) is getting much attention for its therapeutic potential against general ALD and severe AH [172,173]. Given the close association between the gut and liver in the pathogenesis and disease progression of ALD, targeting the microbiome has emerged as a promising approach to treat ALD.

#### 4.4.3. G Protein-Coupled Receptors (GPCRs)

During the course of ALD progression, a variety of cell-extrinsic signaling (e.g., death receptor ligands, DAMPs, and other inflammatory cytokines) can further potentiate alcohol-induced liver injury, inflammation, and fibrosis via receptor-mediated signaling on the cell membrane. Among the broad range of receptor families, G protein-coupled receptors (GPCRs), identified as having over 800 individual encoding genes in the human genome so far, represent one of the largest superfamilies of proteins that serve as a central hub transducing extracellular stimuli to intracellular signaling pathways [174,175,176]. It is now well established that GPCRs can be important pharmacological targets partly due to their feature of membrane-spanning domains which may provide highly accessible sites, thus druggable, at the cell surface [174,175,176]. A recent analysis further supports their clinical relevance that ~34% of all drugs approved by the US Food and Drug Administration (FDA) and ~20% of yet unapproved drugs that are currently in clinical trials target GPCRs [177]. Given their widespread as well as distinct expression pattern across various tissues and cell types, GPCRs regulate a plethora of physiological processes and thus have been implicated in many diseases [175,176,177]. Notably, many drugs targeting GPCRs have been evaluated for their therapeutic potential against metabolic diseases such as type 2 diabetes, obesity, and cardiovascular diseases [178]. It is not surprising that many GPCR-targeting drugs are under investigation or in clinical trials for the treatment of NAFLD/NASH since they are critically involved in the early pathogenesis of various metabolic diseases [179]. According to recent analysis, it has been reported that over 30 GPCRs have been identified as being involved in NASH progression [179]. Given that ALD shares many pathophysiological features with NAFLD during liver disease progression (Table 1), it is thus highly plausible to speculate that GPCR signaling pathways are either directly or indirectly involved in the pathogenesis of ALD and may also serve as promising therapeutic targets. A recent comprehensive analysis of paired liver-plasma proteomes from a large cohort of patients with a diverse spectrum of ALD also showed that over 20% of proteins that were significantly increased in ALD belong to GPCR signaling, thereby demonstrating the potential of the GPCR signaling pathway as a novel therapeutic target against ALD [180]. Herein, we briefly summarize several potential approaches targeting GPCR and associated signaling pathways for treating ALD.

Mounting evidence has revealed that the endogenous cannabinoid system plays a significant role in not only the brain or central nervous system but also peripheral tissues including the liver [181,182]. Both endogenous and exogenous cannabinoids bind to two representative receptors, namely cannabinoid receptor type 1 (CB1R) and type 2 (CB2R) that are both coupled with Gi/o protein, though the expression pattern, as well as the abundance of those receptors in peripheral tissues quite varying depending on cell types [182,183,184]. Following the identification of CB1R and, to a lesser extent, CB2R in the liver, numerous studies have demonstrated that the endocannabinoid system plays a crucial role in the progression of chronic liver disease such as NAFLD, liver fibrosis, and ALD [181,182,185]. Chronic alcohol exposure significantly upregulated hepatic CB1R expression and endocannabinoid level in mice, favoring the development of hepatic steatosis [186]. In addition, CB1R activation by alcohol markedly upregulated the gene expression of estrogen-receptor-related gamma (ERRγ), which subsequently caused CYP2E1 gene induction and thus increased ROS-mediated liver injury [141]. Moreover, global or hepatocyte-specific CB1R gene ablation dramatically suppressed alcohol-induced fat accumulation and oxidative injury in the liver [141,186], indicating the key role of hepatic CB1R in ALD. In contrast, given the predominant expression of CB2R in immune cells within the liver, selective CB2R gene knockout in macrophages (i.e., Kupffer cell) markedly exacerbated ALD by modulating Kupffer cell polarization and autophagy-dependent pathway [187,188]. Thus, raising the necessity of fine-tuning the balance between CB1R and CB2R signaling in the context of ALD treatment. A recent study also demonstrated that the expression of metabotropic glutamate receptor-5 (mGluR5), another GPCR coupled to Gq protein, is increased by alcohol intake in hepatic stellate cells which triggers 2-arachidonoylglycerol (2-AG) production. Since hepatic stellate cell-derived 2-AG binds CB1R to promote de novo lipogenesis, targeting mGlu5 in hepatic stellate cells can be another interesting therapeutic intervention attenuating ALD progression [189].

Chemokine receptors also belong to the GPCR superfamily that regulates various cellular processes in conjunction with immune cell infiltration in response to various stimuli. In particular, aberrant activation of several chemokine–chemokine receptors has been implicated in acute and chronic liver diseases including alcoholic hepatitis, and are therefore regarded as promising pharmacologically targetable molecules [190,191]. Many of chemokine signaling receptors have been identified as being involved in ALD pathogenesis including C-C chemokine receptor type 2 (CCR2), CCR5, CCR6, CXCR1/2 and their cognate chemokines [71,191,192,193,194]. Cenicriviroc (CVC), a dual CCR2/CCR5 antagonist with nanomolar efficacy for both receptors, has recently received considerable attention for its anti-inflammatory and anti-fibrotic efficacy [194,195,196,197]. CVC has recently undergone clinical trials in subjects with NASH and liver fibrosis (CENTAUR study) and has shown a marked improvement in terms of fibrosis without worsening steatohepatitis [194]. However, subsequent clinical trials (AURORA study) were temporarily terminated based on the result of planned interim analysis (ClinicalTrials.gov Identifier NCT03028740 and NCT03059446), possibly due to the nature of chemokines since one chemokine generally targets multiple receptors in the setting of inflammatory disease [190,198]. Nevertheless, considering its therapeutic potential having been evaluated in the rodent ALD model, CVC would still be an attractive target molecule for the treatment of ALD [194]. Similarly, CXCR1/2, namely IL-8 receptors that belong to GPCRs, also play an important role in neutrophil recruitment and activation. A recent study revealed that the level of circulating IL- 8 was observed and may account for AH progression by mediating neutrophil recruitment. The administration of a short lipopeptide (i.e., pepducin ′x1/2pal-i1′) that targets CXCR1/2 markedly attenuated fully established ALD in mice [193], thus showing the potential of CXCR1/2 blockade for the treatment of ALD. Likewise, the gene expression and serum levels of CCL20, the only chemokine ligand of CCR6, strongly correlated with the severity of AH and gene silencing, significantly ameliorated lipopolysaccharide (LPS)-induced inflammation and liver injury [192,199]. Given the contribution of chemokine–chemokine receptors toward the inflammatory milieu, numerous other chemokines may also participate in the pathogenesis of ALD including CXCL5, CXCL6, Gro-α, and IL-18, establishing their potential value as new biomarkers and/or therapeutic targets for ALD [11,20,191,200]. Based on recent findings mostly from pre-clinical studies, other GPCRs that might possess potential therapeutic options against ALD may include bile acid-receptor TGR5 [201,202], adenosine receptors [203,204,205,206,207], and some purinergic receptors [208,209,210]. However, the therapeutic efficacy of those tentative targets may need further in-depth validation to enter clinical trials.

## 5. Conclusions and Perspectives

ALD has been regarded as one of the most prevalent causes of chronic liver disease-related mortality worldwide [211] and is also the leading indication of liver transplantation [212]. Although research has been on-going for several decades to identify novel therapeutic targets for ALD, no medication has been approved by the FDA for the treatment of ALD. This limitation may be partly attributed to the discrepancy between the basic and clinical research modalities in the context of modeling the spectrum of ALD pathophysiology since there are still no adequate experimental ALD models that can fully reproduce the key characteristics of ALD in humans [198]. Therefore, the identification of novel therapeutic targets and development of new therapies are urgently required.

As a continuing effort to overcome the efficacy gap between preclinical and clinical outcomes and solve this health issue, numerous molecular targets within the course of ALD pathogenesis have been studied and some of them are being used or under clinical trials for evaluating their therapeutic efficacy with respect to general pathologic features of ALD such as hepatic steatosis, inflammation, oxidative stress-mediated cell death, and fibrosis. Importantly, it should be noted that fatty liver serves as a prerequisite following similar mechanisms in developing hepatic inflammation in both ALD and NAFLD, as it triggers their progression toward a severe form of chronic liver disease. Thus, it can be inferred that currently explored or future candidate molecular targets for the treatment of NAFLD can also be applied for the treatment of ALD in order to validate their therapeutic potential.

In this review, we described the cellular and molecular pathogenesis of ALD and summarized not only currently available therapies but also some candidate targets that may exhibit promising efficacies for the cure of ALD. Further, among the various possible signaling pathways, GPCR may particularly provide an important and advantageous option as a novel target for ALD treatment, based on its broad diversity and impact on a myriad of biological processes, wide distribution range across most tissues and cell types, and its accessibility to the target on the cell membrane by drugs. As some clinical trials of several drugs targeting the GPCR system have been completed or are currently being undertaken for NAFLD or NASH (e.g., cenicriviroc and namodenoson, see Table 2), it is highly presumable that some of those drugs may be worth evaluating for their possible efficacy and underlying molecular mechanisms for the treatment of ALD. However, it is noteworthy that we generally tend to study and evaluate the role of GPCRs individually, not considering the nature of the GPCR system in that several GPCRs usually form heteromers in response to ligand binding, such that the GPCR ligand can activate multiple GPCR. Given the complex pathophysiology and complicated interaction between the extracellular environment (e.g., PAMPs, DAMPs, inflammatory cytokines, and miscellaneous signaling molecules) and intracellular signaling throughout the ALD progression, future studies should aim at identifying novel signal sensor/transducer molecules (i.e., GPCR system) that can regulate the expression and activity of intracellular molecules responsible for ALD pathogenesis. This may provide additional therapeutic options for the development of a novel therapy against ALD.

## Figures and Tables

**Table 1 biomedicines-10-02530-t001:** The similarities and differences in cellular response and affected signaling pathways between ALD and NAFLD during disease progression.

Similarities: Shared Signaling Pathways/Molecules between ALD and NAFLD			
Cellular Response	Affected Signaling Molecules/Pathways		References
Hepatic fat accumulation	Increased: SREBP1c, SIRT1, ACC, SCD, DGAT Decreased: PPARs, AMPK, CPT		[42]
Cell death	Increased: TRAIL-R2, cell-intrinsic organelle stress, TNFα		[42,70,71,72,73]
Fibrogenesis	Increased: MCP1/CCR2, HMGB1/TLR4, TGFβ		[42]
Several forms of hepatocyte death	Increased: apoptosis, necroptosis, pyroptosis, ferroptosis		[74]
miRNA	Increased: miR-155 Decreased: miR-122, miR-320, miR-486, miR-705, miR-1224, miR-27b, miR-214, miR-199a, miR-192, and miR-183		[42,75]
**Differences: Distinct Signaling Pathways/Molecules between ALD and NAFLD**			
Cellular Response	Affected Signaling Molecules/Pathways		References
	ALD	NAFLD	
Hepatic fat accumulation	Microsteatosis > Macrosteatosis	Microsteatosis < Macrosteatosis	[76]
Lipotoxicity	The contribution of lipotoxiciy not clearly defined	Increased	[42,77]
Insulin resistance	Often accompanied but not clearly defined	Frequently involved and associated with hyperglycemia and type 2 diabetes	[42,78]
Inflammation (macrophage)	Myd88 not involved in TLR4 signaling: type 1 IFNs	Myd88 recruited to TLR4 : proinflammatory cytokines	[42,53,64,79,80]
Microbiota (in fecal)	Increased: Candida albicans, endothelin-converting enzyme	Increased: Prevotella, Porphyromonas Decreased: Bactroidetes	[42,81,82,83]
miRNA	Increased: miR-217, miR-132	Increased: miR-34a Decreased: let7d	[42,84,85,86,87,88,89]
Inflammasome activation	Mostly observed in macrophages during early phase of disease progression	Occasionally observed in hepatocytes during late phase of disease progression	[42]
Necroptosis	RIPK-3 dependent RIPK-1 independent ALT < AST	RIPK-3 independent ALT > AST	[42,57,90]

**Table 2 biomedicines-10-02530-t002:** Recent therapeutic molecules and related clinical trials for the treatment of ALD/AH and NAFLD/NASH.

Agents	Classification (Type of Molecule)	Mechanism of Action	Clinical Trials (Identifier)	
			ALD/AH	NAFLD/NASH
Anakinra (+Zinc)	Anti-inflammation: Lowers hepatic inflammation (IL-1 receptor antagonist)		NCT04072822	
			NCT01809132	
			NCT03775109	
Metadoxine	Antioxidant: Hepatoprotection from oxidative stress (alcohol metabolism inducer)		NCT02161653	NCT02051842 NCT02541045
			NCT02019056	
			NCT01504295	
Emricasan (IDN-6556)	Hepatocellular protection: Lowers apoptosis (pan-caspase inhibitor)		NCT01912404 NCT01937130	NCT02077374
				NCT02686762
				NCT02960204
Selonsertib (GS-4997)	Hepatocellular protection: Lowers apoptosis (ASK1 inhibitor)		NCT02854631	NCT03053050
				NCT03053063
				NCT03449446
				NCT02781584
G-CSF	Hepatocellular regeneration: Promotes liver regeneration (Growth factor		NCT04066179	
			NCT01820208	
			NCT03703674	
IL-22 (F-652)	Hepatocellular regeneration: Lowers inflammation and increases liver regeneration (IL-10 family cytokine)		NCT01918462	
			NCT02655510	
Obeticholic acid	Hepatocellular regeneration and protection: increases liver regeneration and improves cell viability and cholestasis (FXR agonist)		NCT02039219	NCT03836937
				NCT01265498
				NCT02548351
				NCT02633956
Amoxicillin clavulanate	Anti-infection and anti-inflammation: decreases the risk of infection and lowers inflammation (Antibiotics)		NCT02281929	
Rifaximin	Anti-infection and anti-inflammation: decreases the risk of infection and lowers inflammation (Antibiotics)		NCT02116556 NCT02485106	NCT02884037
				NCT01355575
				NCT02009592
Cenicriviroc	Anti-inflammation: Lowers hepatic inflammation (CCR2/CCR5 dual inhibitor)			NCT03028740
				NCT02217475
				NCT03059446
				NCT03517540
Namodenoson	Anti-inflammation and anti-steatosis: Lowers hepatic inflammation and hepatic fat contents (Adenosine A3 receptor agonist)			NCT04697810
				NCT02927314

## Data Availability

Not applicable.

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
