# Peer review of "Alcohol-Related Liver Disease: An Overview on Pathophysiology, Diagnosis and Therapeutic Perspectives"

_biomedicines, 2022, doi:10.3390/biomedicines10102530_

Round 1

Reviewer 1 Report

The review is interesting and thoroughly describes its cellular and molecular pathophysiology, and candidate therapeutic options of alcohol-related liver disease. It is well written, and is easy to read. It will be helpful for the readers interested in alcohol-related liver disease.

Author Response

Dear Reviewer:

 Thanks so much for your high evaluation on our manuscript. During our revision, we found that one reference in the section 4.4.3 (G protein-coupled receptors) was mistakenly omitted, which was added in our revised manuscript (ref. 192, highlighted in red in the reference list).

Please note that since the manuscript file which was provided from the Journal did not work properly with my own reference list editing software, we inevitably revised our manuscript using the original file (in a form that was submitted previously). As we did not change anything throughout the manuscript except some new references and two words within the body of the manuscript (all changes are highlighted in red), we hope that you may easily find a couple of modifications in our revised manuscript.

Again, we do appreciate your generous evaluation of our manuscript.

Sincerely yours, 

Tae Hyun Kim, Ph.D. 

Reviewer 2 Report

The manuscript of Yoonji Ha et al. aims to provide a comprehensive review of alcohol-related liver pathologies with a detailed description of current knowledge concerning the underlying molecular/signaling events. In addition, Authors provide a detailed comparison between alcohol-related liver disease (ALD) and non-alcoholic fatty liver disease (NAFLD). Authors also review the currently available therapeutic ALD/NAFLD options and the provide a summary about the ongoing clinical trials. 

Authors complement the manuscript with 2 detailed Tables and cite more than 200 publications including some very recent (2021-2022) studies.

The topic is of current interest and fits the scope of “Biomedicines”. The manuscript meets the quality requirements of the journal and is of interest for its readers. The text is well written, easy to follow and very informative from both the basic science and clinical point of view.

This reviewer notes only these two minor issues that should be addressed by the Authors:

1.     Page 9, Chapter “4.2.2. Oxidative stress”. Authors should mention the existence of NADPH oxidases (NOX-es) and their role in diverse liver pathologies. This reviewer acknowledges that the review is already very detailed thus, suggests the insertion of the following phrase. 

                   “NADPH oxidase enzymes (NOX-es) are important components of the cellular redox homeostasis and their dysregulated ROS production is associated with diverse liver pathologies. The roles of NOX-es in ALD and NAFLD have been extensively reviewed in recent publications (1,2) and thus, are not included in this review.”

 1. Oxidative Stress, Genomic Integrity, and Liver Diseases. Molecules 2022, 27, 3159. https://doi.org/10.3390/molecules27103159

 2. NADPH Oxidases Connecting Fatty Liver Disease, Insulin Resistance and Type 2 Diabetes: Current Knowledge and Therapeutic Outlook. Antioxidants 2022, 11, 1131. https://doi.org/10.3390/antiox11061131

2.     Table 1: Table 1 appears both on page 5 and 14.

Author Response

Dear Reviewer, 

Thanks so much for your high evaluation on our manuscript with valuable comment.

As per your suggestion, we again read our manuscript carefully. We found that the section that you mentioned (i.e., 4.2.2. Oxidative stress) mainly describes the therapeutic options in each paragraph with regard to their corresponding cellular signaling. Thus, to briefly mention the importance of NADPH Oxidase in oxidative stress during ALD progression, we think that the content that you kindly suggested can be placed at the end of "section 2.2.2 (Alcoholic steatohepatitis/alcoholic hepatitis)" as an additional example of ROS generating signaling molecule (The inserted words are highlighted in red and the corresponding references were also marked in red as presented in the reference).

Please note that since the manuscript file which was provided from the Journal did not work properly with my own reference list editing software, we inevitably revised our manuscript using the original file (in a form that was submitted previously). As we did not change anything throughout the manuscript except some new references and two words within the body of the manuscript (all changes are highlighted in red), we hope that you may easily find a couple of modifications in our revised manuscript.  

Again, we do appreciate your kind suggestion which may greatly improve our manuscript.

Best regards,

Tae Hyun Kim, Ph.D.